# Ag$_3$PO$_4$ enables the generation of long-lived radical cations for visible light-driven [2 + 2] and [4 + 2] pericyclic reactions

Lirong Guo[1], Rongchen Chu[1], Xinyu Hao[1], Yu Lei[2], Haibin Li[1], Dongge Ma [3], Guo Wang[4], Chen-Ho Tung[1] & Yifeng Wang [1] ✉

Photocatalytic redox reactions are important for synthesizing fine chemicals from olefins, but the limited lifetime of radical cation intermediates severely restricts semiconductor photocatalysis efficiency. Here, we report that Ag$_3$PO$_4$ can efficiently catalyze intramolecular and intermolecular [2 + 2] and Diels-Alder cycloadditions under visible-light irradiation. The approach is additive-free, catalyst-recyclable. Mechanistic studies indicate that visible-light irradiation on Ag$_3$PO$_4$ generates holes with high oxidation power, which oxidize aromatic alkene adsorbates into radical cations. In photoreduced Ag$_3$PO$_4$, the conduction band electron ($e_{CB}^-$) has low reduction power due to the delocalization among the Ag$^+$-lattices, while the particle surfaces have a strong electrostatic interaction with the radical cations, which considerably stabilize the radical cations against recombination with $e_{CB}^-$. The radical cation on the particle's surfaces has a lifetime of more than 2 ms, 75 times longer than homogeneous systems. Our findings highlight the effectiveness of inorganic semiconductors for challenging radical cation-mediated synthesis driven by sunlight.

Aromatic alkene radical cations, which are one electron (1e)-oxidation intermediates of aromatic alkenes, play important roles in the synthesis of complex functionalized molecules and cyclic moieties, particularly the [2 + 2] and [4 + 2] pericyclic products. To generate and make use of aromatic alkene radical cations, extensive efforts have been devoted for a long time. Compared to single electron oxidants such as Ce$^{4+1}$, Fe$^{3+2}$, and hypervalent iodine reagents[3,4], photocatalysts (PCs) generate highly oxidizing holes under sunlight irradiation and operate under mild conditions, making photocatalysis a green and sustainable strategy for radical cation-mediated reactions[5–16]. However, the PCs utilized for generating radical cations are primarily homogeneous organic compounds, particularly transition metal-coordination complexes[8–11] and π-conjugated molecules[12–15]. Meanwhile, the scope of the approaches is constrained by the short lifetime

of radical cations (on a μs scale[17–23]). In contrast, inorganic semiconductor PCs (isPCs), such as TiO$_2$, CdS, Bi$_2$MoO$_6$, and Ag$_3$PO$_4$, have been widely employed for solar light harvesting applications, including water splitting, organic pollutant degradation, and photoelectric conversion[24]. From a practical standpoint, they are generally considered stable, recyclable, inexpensive, and environmentally friendly, making them an ideal choice for use in photosynthesis[25]. However, the efficiency of isPCs in the 1e-oxidative activation of non-polar and non-coordinative C = C moieties on their surfaces is typically low[17,20,26–33]. One obstacle is the short lifetime of holes (fs to ns[34]), which significantly slows down the 1e-oxidation of the C = C moieties (Fig. 1a). Moreover, even if some alkene radical cations are slowly generated at the surfaces of a traditional isPC like TiO$_2$, they undergo few intermolecular C-C formation reactions[17]. Instead, they are more prone to

[1]Key Lab for Colloid and Interface Science of Ministry of Education, School of Chemistry and Chemical Engineering Shandong University Jinan, 250100 Jinan, China. [2]Key Laboratory of Photochemistry, Institute of Chemistry Chinese Academy of Sciences Beijing, 100190 Beijing, China. [3]College of Chemistry and Materials Engineering Beijing Technology and Business University Beijing, 100048 Beijing, China. [4]Department of Chemistry Capital Normal University Beijing, 100048 Beijing, China. ✉e-mail: yifeng@sdu.edu.cn

## a

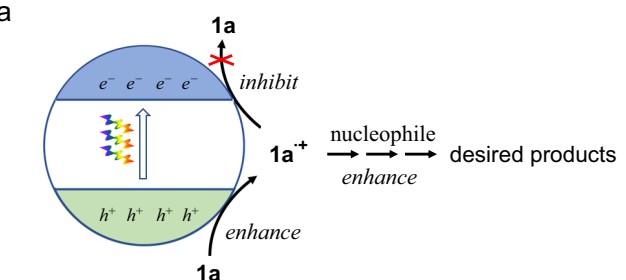

## b

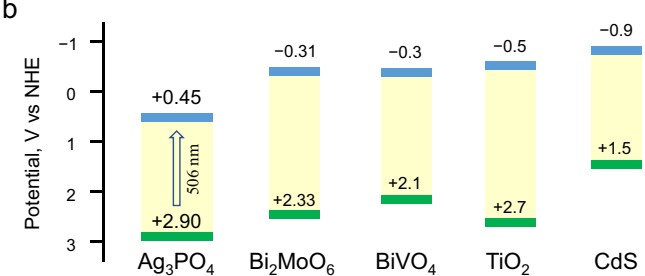

**Fig. 1 | The selection of an inorganic semiconductor for photocatalytic pericyclic reactions. a** Requirements for using anethole radical cation (**1a**•+) in photocatalytic constructing functionalized molecules from anethole (**1a**). **b** The band positions of Ag₃PO₄ and commonly applied photocatalysts.

nucleophilic attack by long-lived photogenerated electrons ($e_{CB}^-$), resulting in ineffective 1e-oxidation. These hindrances lead to a diminished quantum yield of light (i.e. reduced reaction rate) and low product yield, regardless of reaction time. Therefore, it is imperative to extend the lifespan of alkene radical cations for affordable and recyclable PCs that can effectively harvest sunlight for pericyclic reactions. However, this remains an ongoing challenge.

Among all the available isPCs, Ag₃PO₄ stands out as one of the few that exhibits an inherent visible-light response[35]. It has been extensively used in visible-light-driven water oxidation[35–37] and organic pollutant degradation[37,38]. Compared to other commonly used semiconductors, it produces strong oxidizing holes and weak reducing electrons under visible-light irradiation (+2.9 V and +0.45 V *vs.* NHE; Fig. 1b). Moreover, the surfaces of Ag₃PO₄ are rich in large PO₄³⁻ anions with high charge density that can strongly electrostatically interact with cationic species such as radical cations. As a result, photo-excited Ag₃PO₄ may efficiently generate but inefficiently quench radical cations, allowing for the accumulation of radical cations to facilitate radical cation-mediated reactions.

In this work, we demonstrate that Ag₃PO₄ efficiently catalyzes intramolecular and intermolecular [2 + 2] and Diels-Alder cycloadditions under visible light or solar irradiation. The system exhibits remarkable efficiency with respect to substrate scope, product yield, diastereoselectivity, apparent quantum yield (AQY), and scaleup synthesis under solar irradiation. Two critical aspects of the reaction mechanism are validated: (1) the existence of long-lived **1a**•+ radical cations on the surfaces of Ag₃PO₄ that are photo-reduced in situ, (2) the acceleration of rate-limiting step by prolonging the lifetime of **1a**•+. Our discoveries may pave the way for employing highly active organic radical cations and even radical anions in critical pericyclic processes.

## Results

### Transformation of aromatic alkene 1a to *anti*-cyclobutane 2a

Anethole (**1a**) has been the most studied electron-rich $\beta$-substituted styrene in [2 + 2] cycloaddition reactions[3,19,39]. Therefore, it was selected as the model compound to evaluate the photocatalytic performance of Ag₃PO₄ (Fig. 2). A self-synthesized Ag₃PO₄ sample composed

**Fig. 2 | The model reaction of 1a.** Conditions: **1a**, 1 mmol; Ag₃PO₄, (27 mg, 0.12 equiv); HFIP, 3.0 mL; LED (425 nm); 1 atm N₂; 0 °C; 12 h.

of nanospheres with a diameter of 230 ± 60 nm was used for the study (Fig. 3a). The cycloaddition reaction was conducted under an N₂ atmosphere by irradiating a suspension of reactant **1a** in hexafluoroisopropanol (HFIP) solvent containing a catalytic quantity of Ag₃PO₄ (9.0 g L⁻¹ dosage, 0.12 equiv) at 0 °C. No additive was introduced. The conversion of **1a** proceeded smoothly, affording *anti*-cyclobutane **2a** as the sole product (see Supplementary Fig. 1 for the time-resolved ¹H NMR spectra and kinetics). After 12 h of reaction, the yield of **2a** reached 82%, with a high diastereoselectivity (*d.r.* > 19:1). Further irradiation did not affect conversion due to equilibration between **1a** and **2a** (Supplementary Fig. 2), as commonly observed in photocatalyzed radical cation processes[39,40]. Due to this equilibrium, the 82% yield was also achieved using various Ag₃PO₄ samples, including a commercially available Ag₃PO₄ and several self-synthesized faceted Ag₃PO₄ nanocrystal samples (see Supplementary Table 1 for screening of the experimental conditions). This implies that the catalyst is easily accessible.

The control experiments demonstrated that in the absence of Ag₃PO₄ or light, no conversion of **1a** occurred (Supplementary Table 1). Therefore, both light and Ag₃PO₄ are indispensable for the reaction, excluding the possibility of a thermocatalytic mechanism. Notably, the [2 + 2] cycloaddition reaction requires an inert atmosphere. Under air atmosphere, the conversion of **1a** could reach 100%, but 4-methoxybenzaldehyde was the main product (82% yield). The yield of the cycloaddition product **2a** was only 13%. This suggests that Ag₃PO₄ facilitates the selective oxidation of aromatic alkenes, resulting in the formation of the corresponding aromatic aldehydes, when O₂ participates in the reaction process. We also performed the reaction in other solvents like tetrahydrofuran (THF), MeOH, EtOAc, and CF₃CH₂OH, but they were not as effective as HFIP. As shown in Fig. 3b, Ag₃PO₄ can effectively harvest visible light up to 500 nm to initiate the **1a** → **2a** cycloaddition, which is close to its absorption edge. The AQY values were determined according to the initial 30-min yields. The action spectrum of AQY matches the UV-vis diffuse-reflectance spectrum of Ag₃PO₄. The UV-vis absorption spectrum of **1a** shows it can only absorb light below 320 nm, and only Ag₃PO₄ can be excited by the light source (Supplementary Fig. 3). Although we were aware that depositing AgNPs on the surfaces of Ag₃PO₄ could enhance its photoabsorption and charge-separation efficiency[41], our experiments using AgNPs-loaded samples with different AgNPs' loadings afforded lower yields than pure Ag₃PO₄. Furthermore, as the loading of AgNPs increased, the final yield of **2a** decreased (Supplementary Figs. 4–6). This suggests that AgNPs are not the photocatalyst and can inhibit the activity of Ag₃PO₄. In all, it can be concluded that Ag₃PO₄ is the true photocatalyst, and all **2a** is the product of Ag₃PO₄ photocatalysis.

Silver salt-based photocatalysts often suffer from photocorrosion[42–45]. However, in the current system, the 0.12 equiv of initially added Ag₃PO₄ successfully worked five consecutive photocatalytic cycles with only a slight decrease in efficiency (Fig. 3c), resulting in a turnover number of 13. Photo-corrosion of Ag₃PO₄ was observed, as evidenced by its significant darkening after five cycles. The TEM image indicates the formation of numerous AgNPs on the surfaces of Ag₃PO₄ (Supplementary Fig. 7). Nevertheless, regeneration of the recycled sample was achieved through a simple immersion in 6.7 mM Na₂HPO₄ aqueous solution and the addition of a drop of 30% H₂O₂[46]. Within minutes, color, microscopic morphology, and photocatalytic performance were fully restored (Supplementary Fig. 7). As

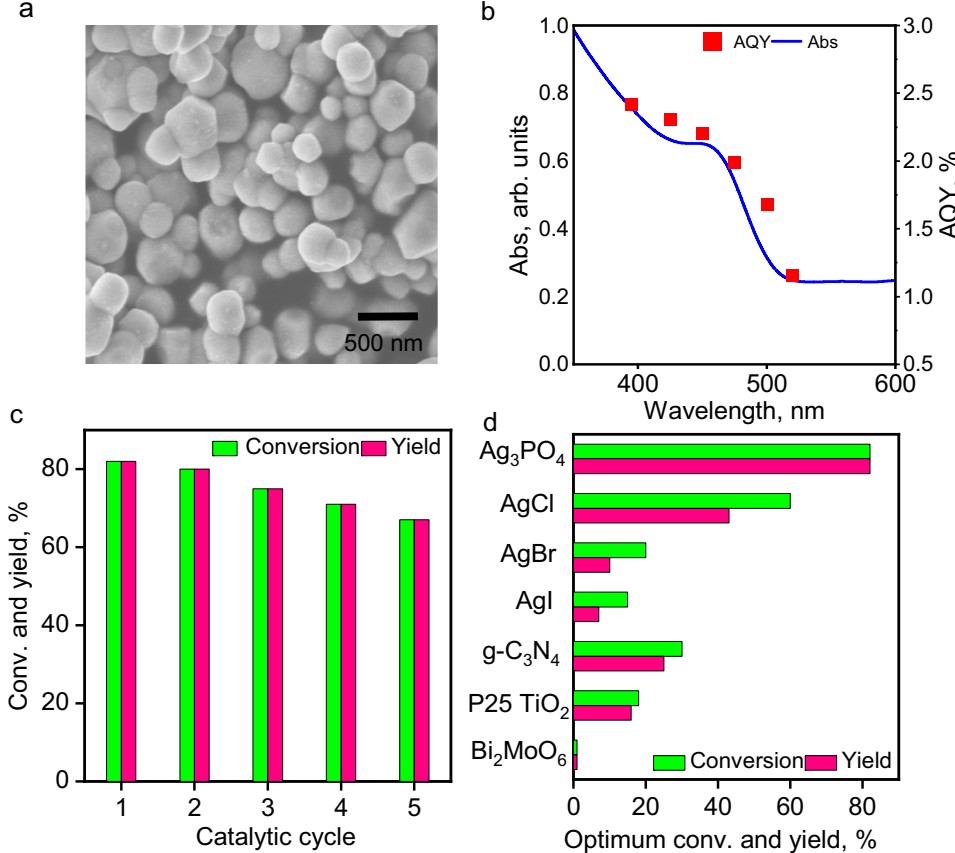

**Fig. 3 | Characterization of Ag₃PO₄ and the yields of 2a under various conditions. a** An SEM image of Ag₃PO₄. **b** The UV-vis diffuse-reflectance spectrum (smooth blue curve) and the AQY action spectrum (red diamond marks; see Supplementary Information for calculation method) of Ag₃PO₄. **c** The conversion of **1a** (green) and yield of **2a** (red) during reuse of Ag₃PO₄. **d** The conversion of **1a** (green) and yield of **2a** (red) by various photocatalysts. The unspecified conditions in panels **b–d**: **1a**, 0.5 mmol; catalyst, 0.12 equiv; HFIP, 1.5 mL; light source, 425 ± 10 nm LED (120 mW cm⁻²); N₂ atmosphere; 0 °C; 12 h. In panel **d**, a 395 ± 10 nm LED lamp (119 mW cm⁻²) was used for TiO₂. The reaction time was optimized for each photocatalyst, reaching up to 24 h for TiO₂. The yields were determined by ¹H NMR using 4-ethoxybenzaldehyde as the internal standard.

shown in Fig. 3d, when the other silver salts, such as AgCl, AgBr, and AgI, were used, the conversion of **1a** and the selectivity of **2a** were significantly lower than that of Ag₃PO₄. Both commercially available and self-synthesized AgCl, AgBr, and AgI decomposed rapidly upon irradiation (see Supplementary Fig. 8 for characterization). In comparison to the widely studied heterogeneous photocatalysts[47], including graphitic carbon nitride (g-C₃N₄), Bi₂MoO₆, and TiO₂, the performance of Ag₃PO₄ is superior. For example, the reaction rate was notably low over TiO₂ even under UV. Furthermore, electron accumulation caused TiO₂ to exhibit a blue hue and ultimately became completely inert once conversion reached 18% (Supplementary Fig. 9). When TiO₂ was used under air atmosphere, the accumulated electrons can be efficiently quenched by O₂. However, in this case the yield of **2a** was less than 10% due to the over-oxidization of **1a** to 4-methoxybenzaldehyde (67%). Supplementary Table 2 shows that Ag₃PO₄ performs comparably to state-of-the-art PCs and single-electron oxidants such as Ru(bpm)₃(BArF)₂[39], PhI(OAc)₂[3] and Fe(ClO₄)₃[19] in the **1a** → **2a** cycloaddition[19]. Therefore, Ag₃PO₄ is highly applicable for [2 + 2] cyclobutanation reactions.

## Ag₃PO₄/visible light system for pericyclic reactions

Given the success of the [2 + 2] homo-cycloaddition of **1a**, we sought to explore whether the Ag₃PO₄/visible light system could serve as a versatile tool for achieving various pericyclic reactions. Our initial focus was on investigating the scope of the intermolecular dimerization reaction. As shown in Fig. 4, a diverse range of electron-rich aromatic alkenes with varying substituents underwent smooth reaction, affording the corresponding symmetrical cyclobutanes in moderate to good yields and excellent diastereoselectivity (**2a**-**2h**, yield ranges 42%-83%, d.r. > 19:1). Electronic density of the aromatic ring and steric hindrance at the β-site exert a noticeable influence on the reaction outcome (**2b** vs. **2c**, **2a** vs. **2d**-**2f**). However, this catalytic system exhibits high tolerance toward steric hindrance at the benzene ring (**2a** vs. **2f**-**2h**). Notably, NO₂-, OH-, and COOH-substituted aromatic alkenes were challenging substrates for other [2 + 2] cycloaddition catalytic systems[48], as well as the current Ag₃PO₄ system (Supplementary Table 3). Finally, we achieved a 70% isolated yield of magnosalin (**2h**), a valuable natural product. We then used the system to synthesize unsymmetric cyclobutanes and were pleased to discover that a variety of aromatic alkenes, including unsubstituted aromatic alkene (**4a**) and substituted aromatic alkenes with electron-donating groups (EDGs; **4b**, **4d**, **4i**) and electron-withdrawing groups (EWGs; **4c**, **4e**, **4f**, **4g**, **4h**) at both ortho-, meta- and para-sites of the benzene rings, could be employed as model reaction counterparts using **1a**. The corresponding intermolecular [2 + 2] products (**2a**−**2h** and **4a**−**4i**) were obtained in high yields and high regioselectivity (head-to-head; see Supplementary Table 4 for ¹H NMR analysis). The sole byproduct resulting from these crossed reactions is the homo-cycloaddition product, e.g., **2a**. However, because the crossed reactions are much faster than the homo-cycloaddition (see Supplementary Fig. 10 for a comparison of the rates), introducing **1a** via a syringe pump would inhibit the homo [2 + 2] reaction. Next, we explored the feasibility of intramolecular

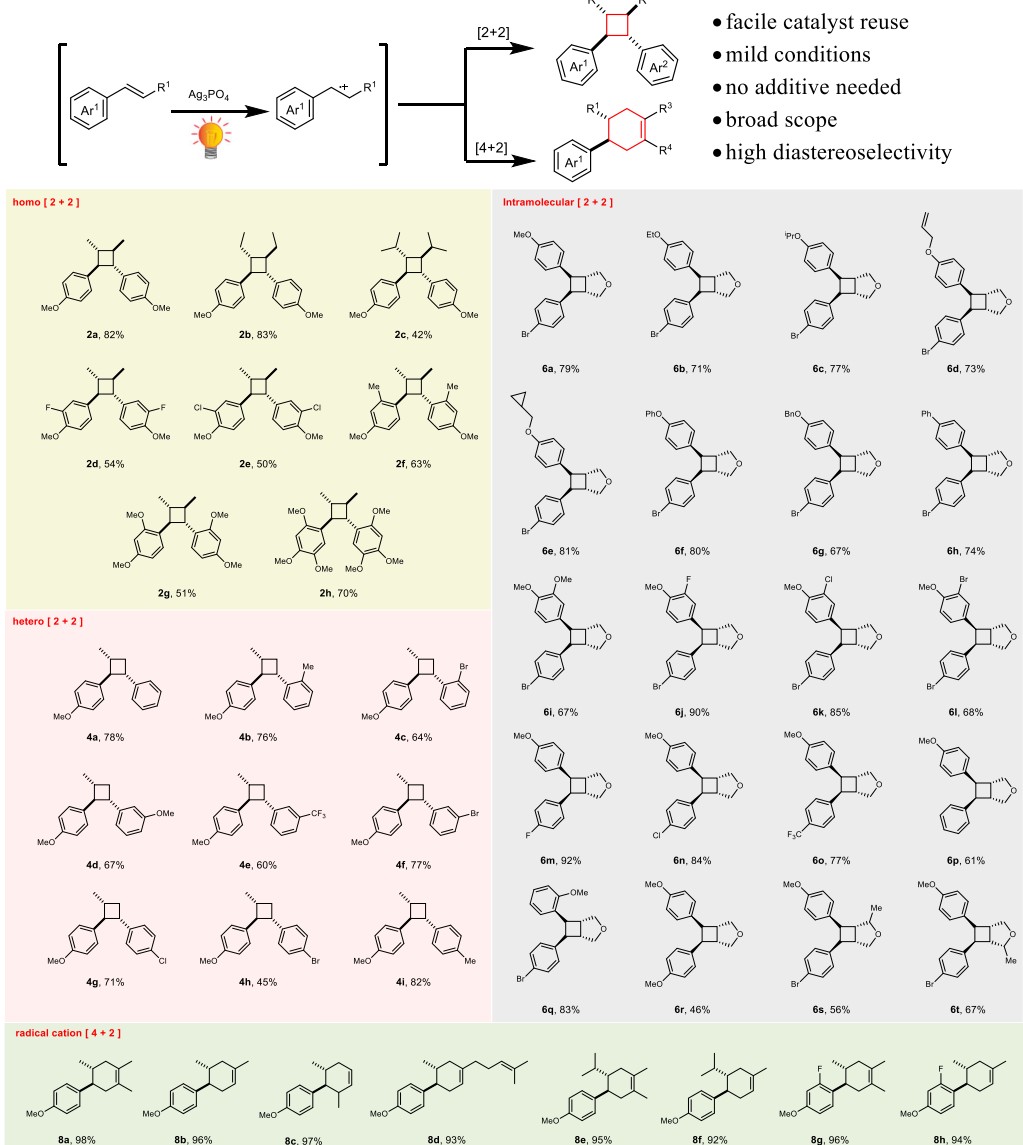

**Fig. 4 | Scope of the Ag₃PO₄ photocatalysis in the radical cation pericyclic reactions.** [a]Conditions for homo [2 + 2] reactions: substrate, 1.0 mmol; Ag₃PO₄, 27 mg (0.12 equiv); 12–18 h. [b]Conditions for hetero [2 + 2] reactions: **1a**, 0.5 mmol; the counterpart, 1.0 mmol; Ag₃PO₄, 44 mg (0.20 equiv); 12 h. [c]Conditions for intramolecular [2 + 2] reactions: substrate, 0.3 mmol; Ag₃PO₄, 12 mg (0.10 equiv); 12 h. [d]Conditions for [4 + 2] reactions: aromatic alkene, 1.0 mmol; diene, 2.0 mmol; Ag₃PO₄, 32 mg (0.08 equiv); 8 h.

[2 + 2] cycloaddition, an efficient approach for synthesizing fused heterocycles[6]. Strikingly, all tested bis(styrene)s afforded good to excellent yields of the desired cyclobutanes (**6a-6t**). The intramolecular [2 + 2] cycloadditions exhibit a broad tolerance toward EDGs, EWGs, and the steric hindrance at both the benzene ring and the β-site. The Diels-Alder reaction is considered one of the essential C-C bond-forming reactions in synthetic organic chemistry. Recent studies have shown that photocatalytic methods are effective in the radical-cation-mediated cycloadditions of electron-rich olefins and dienes[32,49–52], which are challenging substrates for conventional thermal processes. In this study, we found that even with a reduced Ag₃PO₄ loading of 0.08 equiv, the Ag₃PO₄/visible light system was still capable of facilitating Diels-Alder cycloadditions involving the radical cations. Moreover, the 0.08 equiv of Ag₃PO₄ was successfully reused for five consecutive runs without any noticeable decrease in performance (Supplementary Figure 11). Based on the substituted aromatic alkenes, the yields achieved in all tested reactions are near unity (**8a-8h**). The products **8b, 8c, 8d, 8f, and 8h** were isolated in high regioselectivity

without isomers. This can be attributed to the large steric hindrance of the butadiene side chain (see Supplementary Table 4 for ¹H NMR analysis).

The remarkable diastereoselectivity of the reactions is noteworthy. Specifically, nearly all intermolecular [2 + 2] products are anti; intramolecular [2 + 2] products are predominately syn, and nearly all Diels-Alder reactions yield anti-products.

## Scale synthesis of [2 + 2] and [4 + 2] reactions
To investigate the synthetic potential of the Ag₃PO₄/visible light system, we performed large-scale [2 + 2] and [4 + 2] reactions under natural sunlight irradiation by placing the reaction flasks on a windowsill at outdoor temperatures (1–10 °C). The sunlight intensity ranged from 15–23 mW cm⁻². Interestingly, 41.5 g of the [4 + 2] cycloaddition product **8a** was obtained almost quantitatively in a one-pot reaction after only six hours of irradiation (Fig. 5). After being filtered through diatomite, the Ag₃PO₄ solid was removed, resulting in nearly pure **8a** (41.5 g), which confirms the method's flexibility and ease of use. The

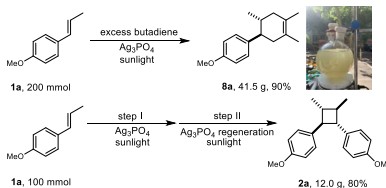

**Fig. 5 | Scaleup synthesis using sunlight.** Detailed reaction conditions were described in the Supplementary Information.

intermolecular [2 + 2] homo-cycloaddition of **1a** yielded a 60% yield of **2a** (100% selectivity) before $Ag_3PO_4$ (0.05 equiv) became deactivated. However, upon reintroduction of regenerated $Ag_3PO_4$ into the catalytic system, a final yield of 80% of **2a** was achieved (Fig. 5). The use of a 425 nm LED lamp achieved yields of **8a** that are comparable to that achieved by sunlight (Supplementary Table 1). Hence, the $Ag_3PO_4$/visible light system could facilitate actual industrial production considering the advances in LED technology.

Supplementary Tables 5–7 demonstrate the superior performance of the $Ag_3PO_4$/visible light system compared to the state-of-art reports in [2 + 2] and Diels-Alder cycloadditions. The current system boasts one of the highest efficiencies in heterogeneous systems, with broad light absorption. It is capable of realizing intramolecular and intermolecular [2 + 2] cycloadditions and Diels-Alder reactions.

## Mechanism study and DFT simulations

Laser flash photolysis (LFP) was performed to detect the transient species involved in the photocatalytic **1a** → **2a** cycloaddition. The LFP transient absorption spectra of the **1a**/$Ag_3PO_4$/HFIP suspension were obtained in transmission mode. The representative spectra are depicted in Fig. 6a. The two distinct, intense peaks centered at ca. 387 and 605 nm precisely match the spectra of **1a**[.+] generated by photolysis of homogeneous solutions of **1a**/$H_2O$-MeCN at 266 nm[53] and **1a**/MeCN at 308 nm[54], respectively, as reported in previous studies. Figure 6b demonstrates that the absence of **1a** or $Ag_3PO_4$ resulted in no signal from a 355-nm laser pulse, indicating the indispensability of both **1a** and $Ag_3PO_4$ for the formation of **1a**[.+]. Hence, the LFP spectra confirm that photogenerated $h^+$ of $Ag_3PO_4$ facilitates the 1e-oxidation of **1a** to afford **1a**[.+].

The concentration of a transient species, which is determined by both formation rate and lifetime, is essential for detecting it via LFP. Hence, the observation of **1a**[.+] in the **1a**/$Ag_3PO_4$/HFIP system suggests that **1a**[.+] forms rapidly and has a long life, allowing for its accumulation due to its formation far exceeding decay. Recall Fig. 6a, the sample initially exhibited strong background absorption, which significantly decreased after 250 μs due to the sedimentation of $Ag_3PO_4$ particles. The spectra at 250 and 500 μs showed similar level of background absorption, indicating that sedimentation was minor during this period. Based on this understanding, we measured the decay kinetics of **1a**[.+] in the **1a**/$Ag_3PO_4$/HFIP suspension at $\lambda = 600$ nm. Figure 6c demonstrates that the signal intensity decayed significantly before 500 μs, which could be mostly attributable to the sedimentation mentioned above of $Ag_3PO_4$. Afterward, the signal intensity remained nearly constant until 1900 μs. This must not be attributed to the continuous generation of **1a**[.+] by the photogenerated holes of $Ag_3PO_4$ after LFP, because it is well-known that the photogenerated holes in semiconductors are transient species. Instead, this indicates that the lifetime of **1a**[.+] in the **1a**/$Ag_3PO_4$/HFIP system is more than 1900 μs.

We conducted light-on-off experiments to investigate the lifetime of **1a**[.+] in the **1a**/$Ag_3PO_4$/HFIP system. The reaction vial was either left stirring in the dark after turning off the light or centrifuged to settle down the $Ag_3PO_4$ before being left in the dark. In the former situation, over the next six hours, there was a gradual increase of ca. 8% in yield of **2a** (Fig. 6d). This is distinct from the homogeneous systems, where the

**1a** → **2a** cycloaddition practically ceased upon light-off[55]. In the dark reaction without agitation, the yield of **2a** remained unchanged after $Ag_3PO_4$ settled down (Fig. 6d). It indicates that the long-lived **1a**[.+] radical cations must remain adsorbed on the surfaces of the reduced $Ag_3PO_4$ (denoted as $(Ag_3PO_4)^{n-}$, where $n$ is the number of $e_{CB}^-$ per NP). When all the $Ag_3PO_4$ particles settled down, the **1a** molecules in solution could not reach **1a**[.+], and thus the cycloaddition ceased. This verifies that the signals originate from **1a**[.+] adsorbed on the $Ag_3PO_4$ surfaces rather than in the solution bulk. Further, it confirms that the radical cation-mediated [2 + 2] pericyclic reaction occurs on the surfaces of $Ag_3PO_4$ rather than in the solution.

It was reported that the decay rate constant[53,56] of **1a**[.+] is $4 \times 10^4 s^{-1}$ in aerated MeCN, corresponding to a very short lifetime of 25 μs[54]. This is likely why the **1a** → **2a** cycloaddition stopped nearly instantly upon light-off in homogeneous systems[22,55]. To probe the lifetime of **1a**[.+] in neat HFIP, we used a 266-nm laser pulse to excite a **1a**/HFIP solution. However, the system produced signals from unknown species and the absorption peaks of **1a**[.+] at ca. 387 and 605 nm were not detected. This may suggest that the lifetime of **1a**[.+] in HFIP is too short, resulting in a **1a**[.+] concentration below the detection limit. We also used a 355-nm laser pulse to excite the **1a**/$TiO_2$/HFIP system, which produced a 16% yield of **2a**, indicating that photo-excited $TiO_2$ NPs generated **1a**[.+]. However, the transient spectrum of **1a**[.+] was not detected by LFP (Supplementary Fig. 12), implying that its lifetime in the system is also very short.

We found that the adsorption of reactant **1a** and desorption of product **2a** are crucial in the reaction. Figure 7a shows that the adsorption of **1a** on the (100)-facet-rich spherical $Ag_3PO_4$ (as the representative example) can fit well to the Langmuir type-I isotherm (Eq. 1),

$$\Gamma = \Gamma_m \cdot K \cdot C_{eq}/(1 + K \cdot C_{eq}) \tag{1}$$

where $\Gamma$ is the adsorbed amount at the adsorption/desorption equilibrium concentration $C_{eq}$. The Langmuir coefficient $K$ and the capacity of adsorption $\Gamma_m$ are calculated to be $5.8 \pm 0.6 M^{-1}$ and $2.9 \pm 0.1$ mmol $g^{-1}$, respectively. In contrast, the adsorption of **2a** on $Ag_3PO_4$ surfaces is barely detectable, indicating very weak adsorption of **2a**.

At low **1a** concentrations, the AQY values increase dramatically with the initially added concentration of **1a** but then approach a plateau (Supplementary Fig. 13). This differs from a typical homogeneous bimolecular reaction, which is second order depending on the substrate concentration. Instead, the AQY values are well linearly correlated with the square of the fractional coverage of **1a**, Θ (Eq. 2),

$$AQY = k(\Gamma/\Gamma_m)^2 = k\,\theta^2 \tag{2}$$

where $k$ denotes the slope of the plot (Fig. 7b). This kinetic equation corroborates that the conversion of **1a** through photocatalysis occurs on the $Ag_3PO_4$ surface. The rate-limiting step (RLS) of the formation of **2a** involves two molecular species of **1a** adsorbed on the surface. We also observe a significant solvent effect in the performance of $Ag_3PO_4$ in the **1a** → **2a** cycloaddition reaction (Fig. 7c). $Ag_3PO_4$ did not perform well in EtOAc, $n$-hexane or $CH_2Cl_2$, but performed better in $CH_3CN$, $MeNO_2$, and TFE, and performed best in HFIP. Surprisingly, Eq. 2 applies to the results from different solvents. Thus, solvents influence the AQY by modulating the **1a** distribution between the bulk solution and $Ag_3PO_4$ surfaces. It has been reported that the facets of photocatalysts substantially influence their activities, especially in the case of $Ag_3PO_4$[36,37,57,58]. For the **1a** → **2a** cycloaddition reaction, we used the spherical, rhombic dodecahedral, cubic, and tetrahedral NPs of $Ag_3PO_4$, which are rich in {100}, {110}, {100}, and {111} facets, respectively[36]. Figure 7d displays distinct AQYs on different facets and Eq. 2 is applicable to the results from various $Ag_3PO_4$ facets. Overall, Fig. 7 indicates that the photocatalytic cycloaddition reaction on

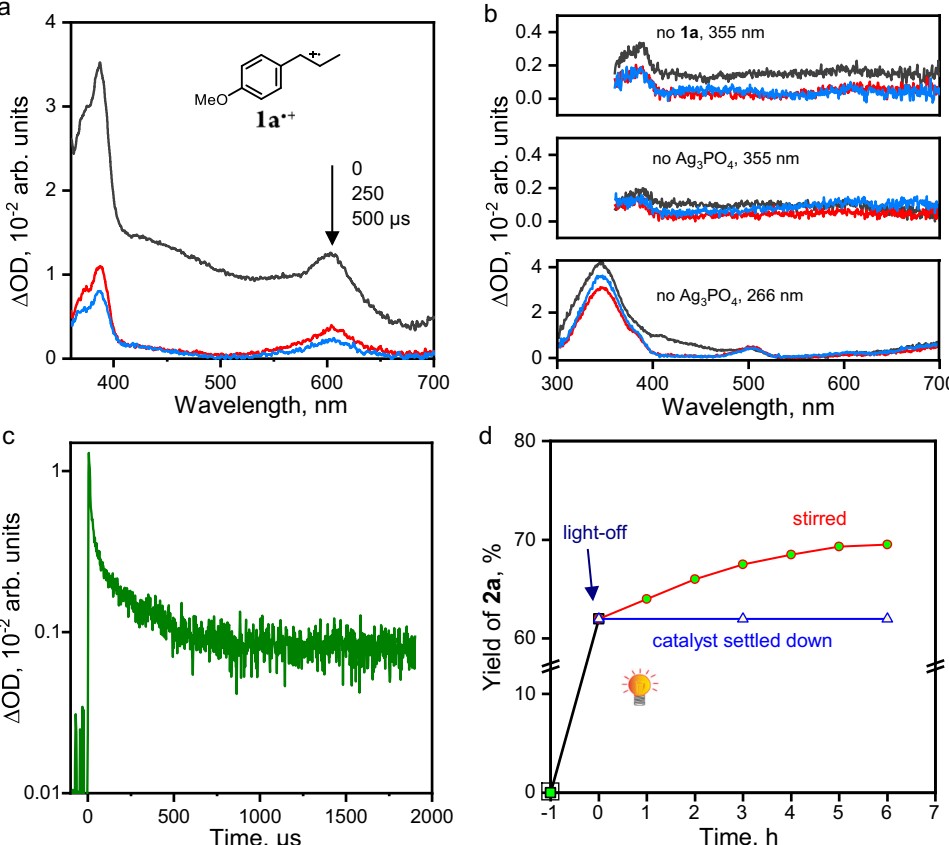

**Fig. 6 | Detection of the transient species. a** Transient absorption spectra of **1a**⁺• obtained at various times after a 10-ns 355-nm pulse irradiation of the **1a**/Ag₃PO₄/ HFIP system at room temperature under an air atmosphere. Before LFP, the suspension was ultrasonicated for 30 min to enhance the dispersion of Ag₃PO₄ in HFIP.

**b** The transient absorption spectra obtained without **1a** or Ag₃PO₄: black, $t = 0$; red, 250 μs; blue, 500 μs. **c** The decay kinetics of **1a**⁺• after 355-nm pulse irradiation. **d** The kinetics of **2a** formation in the light on-off experiment.

Ag₃PO₄ surfaces follows the Langmuir-Hinshelwood mechanism for a bimolecular reaction. It also implies that the interaction between **1a** and Ag₃PO₄ surfaces is vital in the success of Ag₃PO₄ in the photocatalytic [2 + 2] cycloadditions.

We conducted DFT simulations to determine the mechanism of interfacial interactions between **1a** (and **2a**) and the Ag₃PO₄ surfaces. Given the substantial dimensions of the Ag₃PO₄ particles and the comparable sizes of **1a** or **2a** molecules to only a few phosphate anions, it is reasonable to regard the surface of Ag₃PO₄ particles as an infinitely expansive plane for adsorbing **1a** and **2a**. As the (100) facet is the lowest-energy facet of Ag₃PO₄ crystals[36,37,58] and the primary surface of spherical Ag₃PO₄, we selected the PO₄³⁻-terminated and the Ag⁺-terminated (100) facets to calculate the optimal adsorption configurations (see Supplementary Figs. 14–15 for details of results). Figure 8 displays the lowest-energy configurations of **1a** and **2a** molecules on the PO₄³⁻-terminated (100) facet. It should be noted that the size of PO₄³⁻ is much larger than that of Ag⁺ (2.38 Å vs. 0.67 Å), but comparable to molecule **1a** in size. Molecule **1a** lies parallel to the facet, with its long axis aligned with the *a*-axis of the crystal lattice, and the CH = CH moiety positioned close to the O²⁻ ions of PO₄³⁻. In this configuration, each **1a** molecule intimately contacts four PO₄³⁻ anions, maximizing the interfacial interaction between **1a** and the Ag₃PO₄ surface. The strong interaction is consistent with the previously discussed adsorption of **1a** to the Ag₃PO₄ surfaces. This adsorption mode should be beneficial for the 1e-oxidation of the CH = CH moiety upon photo-excitation of Ag₃PO₄ since photogenerated holes are localized at O of PO₄³⁻[59]. The energy required for adsorption of each **1a** molecule from the vacuum is −2.10 eV (Fig. 8a, b). In contrast, the calculated energies for **1a**

adsorption onto PO₄³⁻-terminated (100) facet along the *b*-axis and that onto the Ag⁺-terminated (100) facet along both the *a*- and *b*-axis, are substantially smaller at −1.67, −1.54, and −1.54 eV, respectively (Supplementary Fig. 14). These lower adsorption energies can be attributed to the greater spacing between the adjacent PO₄³⁻ anions on these facets, which weakens the contact between **1a** and PO₄³⁻. However, due to the large steric effects of the **2a** molecule, each **2a** has an optimum adsorption energy of −1.56 eV when adsorbed on the PO₄³⁻-terminated (100) facet (Fig. 8c, d). The above calculations reveal a large adsorption energy difference between two **1a** and one **2a**, i.e. +2.64 eV on the PO₄³⁻-terminated Ag₃PO₄ (100) facet in a vacuum. The much weaker adsorption of **2a** suggests that the **1a**→**2a** cycloaddition on the Ag₃PO₄ surfaces benefits from the easier removal of product **2a**.

The atomic Bader charges shown in Fig. 8e indicate that the electron density of **1a** changes upon adsorption. The overall Bader charge of adsorbed **1a** is 0.34e, indicating a transfer of 0.34e from **1a** to the Ag₃PO₄ surfaces during adsorption. The two H-atoms in the CH = CH moiety have enormous Bader charges, namely, 0.1323e and 0.1e, respectively. This, combined with the very negative Bader charge of the CH₃O moiety (−0.4587e), strongly suggests that polarization occurs in molecule **1a** and that its CH = CH group is activated upon adsorption. The high Bader charge also means that the CH = CH group is the most readily oxidized site in **1a**.

Figure 9 illustrates a plausible mechanism for Ag₃PO₄ triggering the [2 + 2] cycloaddition of **1a** under visible-light irradiation. Initially, Ag₃PO₄ adsorbs **1a** molecules on its surface via PO₄³⁻ anions (step I). Then, upon excitation, Ag₃PO₄ generates many $e_{CB}^-$ in the conduction band and $h^+$ in the valance band (step II). Due to the fact that the

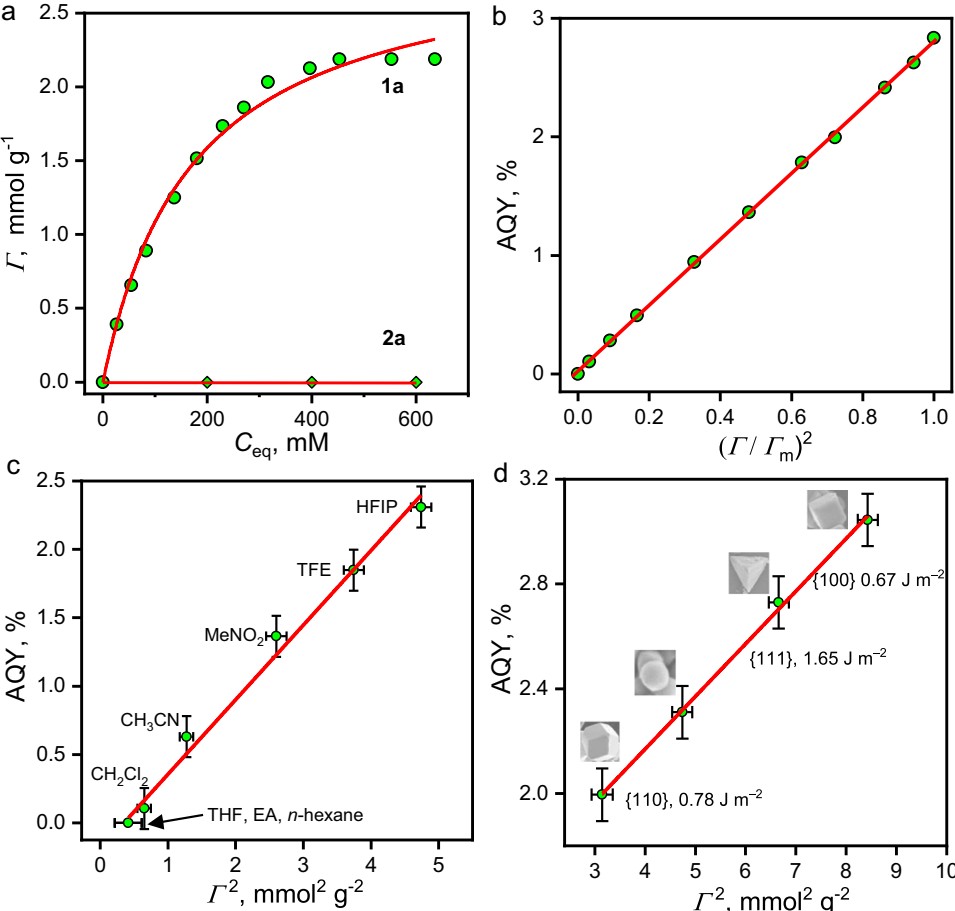

**Fig. 7 | Dependence of AQY on adsorption. a** The adsorption isotherm of **1a** and **2a** on spherical Ag$_3$PO$_4$. **b–d** The AQY values of the **1a** → **2a** cycloaddition reaction as functions of $(\Gamma/\Gamma_m)^2$ or $\Gamma^2$. The scattered points represent the experimental values, and the red lines are the curve fit or linear fits. Abbreviations for panel **c**: THF tetrahydrofuran, EA ethyl acetate, TFE trifluoroethanol. The SEM images, facets, and surface energies of Ag$_3$PO$_4$ NPs are indicated in panel **d**. The error bars in panels (**c**) and (**d**) associated with AQY represent the standard error of three sets of unique measurements.

conduction band bottom is mainly composed of hybridized Ag 5s5p orbitals[35], the reduction power of the electrons in the conduction band of Ag$_3$PO$_4$ is low ($E_{CB}$ = +0.45 V vs. NHE[35,41,60,61]). The unreactive $e_{CB}^-$ remains in the form of $(Ag^{\delta+})_m^-$, which means it is shared by many Ag$^+$ ions. The observed AgNP formation substantiates that $e_{CB}^-$ transfers to Ag$^+$ ions. The valance band top of Ag$_3$PO$_4$ mainly comprises hybridized O 2p and Ag 4d orbitals. The $h^+$ in the valence band has strong oxidation power ($E_{VB}$ = +2.90 V vs. NHE[35,41,60,61]) and is able to gain an electron from the CH = CH group of **1a** to yield **1a**$^{\bullet+}$ (step III). The 1e-oxidation of **1a** results in the photocatalytically reduced NP with numerous $e_{CB}^-$. The **1a**$^{\bullet+}$ species strongly adsorbs on the surfaces of (Ag$_3$PO$_4$)$^{n-}$. In this case, the electrostatic interaction between **1a**$^{\bullet+}$ and (Ag$_3$PO$_4$)$^{n-}$ NP surfaces is analogous to the interactions in homogeneous solutions described by Yoon et al. for **1a**$^{\bullet+}$ with a tetraarylborate anion[18], Ishihara et al. for **1a**$^{\bullet+}$ with FeCl$_4$$^{-19-21}$, and List et al. for **1a**$^{\bullet+}$ with an imidodiphosphorimidate counteranion[22].

The adsorbed **1a**$^{\bullet+}$ is stabilized by electrostatic interaction with the (Ag$_3$PO$_4$)$^{n-}$ surfaces. The low possibility recombination between **1a**$^{\bullet+}$ and $e_{CB}^-$ shared by multiple Ag$^+$ cations, (Ag$^{\delta+}$)$_m^-$, is evidenced by the observation of AgNP formation. As a result, the lifetime of **1a**$^{\bullet+}$ is dramatically prolonged to > 2 ms, which is over 75 times longer than in a homogeneous solution (μs level[54]). The long lifetime of **1a**$^{\bullet+}$ benefits the nucleophilic attack by another adsorbed **1a** molecule, allowing the reaction of **1a** + **1a**$^{\bullet+}$ → **2a**$^{\bullet+}$ in step IV to proceed (the RLS; see Supplementary Information for analysis).

The final step (step V) is the ring closure reaction of **2a**$^{\bullet+}$. The potential of **2a**$^{\bullet+}$/**2a** is approximately +1.5 V vs. NHE, much higher than the $E_{CB}$ of Ag$_3$PO$_4$. Thermodynamically, the transfer of an electron from the reduced Ag$_3$PO$_4$ to **2a**$^{\bullet+}$ is an exothermic reaction. Step V may also occur through a radical chain mechanism[22]. Considering the potential of **1a**$^{\bullet+}$/**1a**, which is approximately +1.3 V vs. NHE, the ring closure of **2a**$^{\bullet+}$ by $e_{CB}^-$ exhibits a much larger driving force than that by the radical chain process.

The ability to generate the long-lived radical cation **1a**$^{\bullet+}$ distinguishes (Ag$_3$PO$_4$)$^{n-}$ from a traditional photocatalytically reduced TiO$_2$ NP. On the illuminated TiO$_2$ NP, the lifetime of **1a**$^{\bullet+}$ was too short to detect, and the conversion was only 18% (Fig. 3d). Okada et al. used a substantial excess of TiO$_2$ (e.g., > 6 equiv) to minimize the accumulated electron per NP and extra LiClO$_4$ (1.0 M) to stabilize the radical cations[18,22,23,25–27]. In contrast, Ag$_3$PO$_4$ photocatalysis can harvest the visible spectrum of sunlight and achieve a high yield of the desired product without any additives.

## Discussion

Ag$_3$PO$_4$ is a powerful photocatalyst for homo, crossed, intramolecular [2 + 2], and Diels-Alder [4 + 2] pericyclic reactions under visible-light irradiation. The catalytic process is mild, straightforward, affordable, additive-free, scalable under sunlight irradiation, and allows for easy product separation and catalyst reuse. It has a broad substrate scope and produces a wide range of desired products in modest to excellent yields. Our study reveals the potential of this photocatalytic process

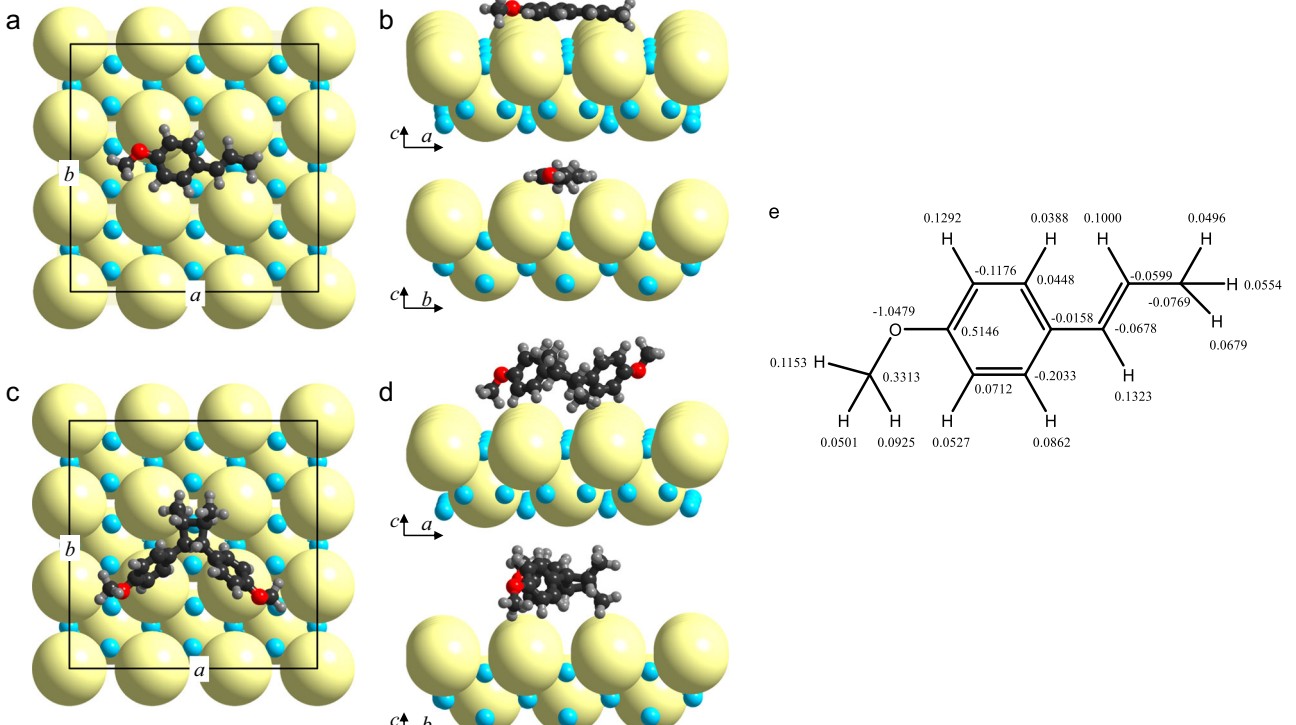

**Fig. 8 | Configurations of adsorption and Bader charge. a–d** The lowest-energy configurations of **1a** and **2a** molecules on the 2 × 2 region of the $PO_4^{3-}$-terminated (100) facet of $Ag_3PO_4$. **a** and **c** are the top views. **b** and **d** are the side views. The labels *a* and *b* denote the cell axes. The pale-yellow spheres represent $PO_4^{3-}$ ($r = 2.38$ Å), and the cyan spheres represent $Ag^+$ ($r = 0.67$ Å). **1a** and **2a** molecules are drawn to scale. **e** The atomic Bader charges of adsorbed **1a**.

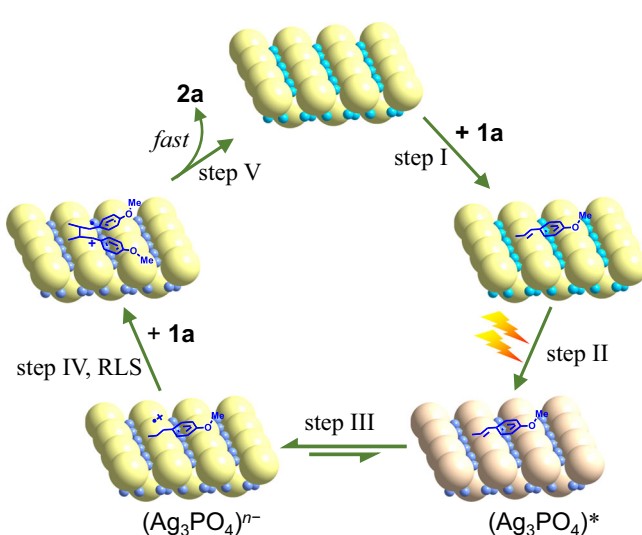

**Fig. 9 | A plausible mechanism.** $(Ag_3PO_4)^*$ denotes a photo-excited $Ag_3PO_4$ NP with many $e_{CB}^-$ and $h^+$. The purple-blue spheres are used to show that in $(Ag_3PO_4)^*$ and $(Ag_3PO_4)^{n-}$, the $e_{CB}^-$ is shared by silver cations, resulting in a fraction charge of $\delta+$ ($0 < \delta < 1$) on each cation.

for fine chemical production. We have demonstrated that the rate-limiting step is the reaction between the reactant and its 1e-oxidation intermediate, a radical cation. The lifetime of anethole radical cation ($1a^{•+}$) on the $Ag_3PO_4$ surfaces exceeds 2 ms, which is over 75 times longer than that in the homogeneous solutions, thus effectively promoting the rate-limiting step. The long lifetime of $1a^{•+}$ is attributed to the appropriate band structure of $Ag_3PO_4$ and the strong electrostatic interaction between $1a^{•+}$ and the $(Ag_3PO_4)^{n-}$ NP surfaces, which should

be a general and essential mechanism for promoting chemical processes mediated by radical cations on heterogeneous surfaces. This may inspire ideas for more challenging radical cation/anion-mediated solar synthesis using inorganic semiconductor photocatalysts.

## Methods

### General procedure for the photocatalytic reactions
The reactions were carried out in 10-mL Pyrex vials. Olefin and $Ag_3PO_4$ were dispersed in a solvent in the vial, which was then purged for 10 min with high-purity $N_2$ (99.999%). The vial was immersed in a mixture of ice and water to maintain the reaction temperature. The reaction suspension was stirred in the dark for 30 min to achieve adsorption-desorption equilibrium. The mixture was then exposed to the lamp from the side. To monitor the reaction progress, a syringe was used to withdraw 10 μL of the solution for analysis by thin-layer chromatography (TLC). After the reaction, the suspension was centrifuged to separate the solid catalyst from the solution. The residue is purified with column chromatography to afford the desired pure products.

### Details for the homo-dimerization of aromatic alkenes
To a solution of aromatic alkene **1** (1.0 mmol) in 3.0 mL of HFIP, $Ag_3PO_4$ (27 mg, 0.12 equiv) was added in one portion. The remaining steps are the same as the general procedure.

### Details for the cross-dimerization of aromatic alkenes
To a solution of aromatic alkene **3** (1.0 mmol, 2.0 equiv) in 2.0 mL of HFIP was added $Ag_3PO_4$ (44 mg, 0.20 equiv). Then, the reaction mixture was degassed by purging with high-purity nitrogen for 10 min. An ice bath was used to maintain the reaction temperature. The mixture was stirred in the dark at 0 °C for half an hour to achieve adsorption-desorption equilibrium. The photocatalytic reaction was then initiated by irradiating the dispersion from the side with the LED lamps. During

the reaction, a solution of **1** (0.5 mmol) in 2.0 mL HFIP was added using a syringe pump (at a rate of 4.0 mL/h). The remaining steps are the same as the general procedure.

### Details for the intramolecular [2 + 2] reactions
To a solution of aromatic alkene **5** (0.3 mmol) in 2.0 mL of HFIP, Ag$_3$PO$_4$ (12 mg, 0.10 equiv) was added in one portion. The remaining steps are the same as the general procedure.

### Details for the Diels−Alder cycloadditions
To a solution of the diene **7** (2.0 mmol, 2.0 equiv) in 2.0 mL of HFIP was added Ag$_3$PO$_4$ (32 mg, 0.08 equiv). Then, the reaction mixture was degassed by purging with high-purity nitrogen for 10 min. An ice bath was used to maintain the reaction temperature. The mixture was then stirred in the dark at 0 °C for half an hour to achieve adsorption-desorption equilibrium. The photocatalytic reaction was initiated by irradiating the dispersion from the side with an LED lamp. During the reaction, a solution of **1** (1.0 mmol) in 2.0 mL of HFIP was added using a syringe pump (at a rate of 4.0 mL/h). The remaining steps are the same as the general procedure.

## Data availability
The authors declare that all relevant data supporting the findings of this study are available either within the manuscript itself and/or in the Supplementary Information. Experimental details and characterization of products are provided in the Supplementary Information. All other data are available from the corresponding author upon request. Source data are provided in this paper.

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

## Acknowledgements

The authors gratefully acknowledge the National Natural Science Foundation of China (Grant Nos 22276112, 21922605, and 22076007), the Natural Science Foundation of Shandong Province (Nos. 2019GSF109065, 2021CXGC011202 and ZR2019ZD45), the Taishan Scholar Project 454 Foundation of Shandong Province (No ts20190908) for the financial supports.

## Author contributions

Y.W. and C.T. conceived the idea and directed the project. L.G. performed the experiments and analyzed the data. R.C., X.H., Y.L., and H.L.; D.M. participated in characterization studies. G.W. performed the DFT calculation. Y.W. and L.G. wrote the manuscript, and L.G. prepared the Supplementary Information.

## Competing interests
The authors declare no competing interests.
