## [Peer Review File · Nature Communications]

Ag₃PO₄ Enables the Generation of Long-lived Radical Cations for Visible Light-Driven [2+2] and [4+2] Pericyclic ReactionsReviewers' Comments:

Reviewer #1:

Remarks to the Author:

In this paper, Wang and co-workers reported a visible light driven [2+2] and [4+2] reactions mediated by Ag₃PO₄ nanoparticles. The reactions show advantages in the mild and additive free reaction conditions, scalable synthesis, and the ease in catalyst regeneration. The paper is well written, and the discussions are reasonable. I have a only a few comments:

1. The effect of temperature and air should be examined more carefully. 0oC was used in the standard condition, but the control experiments by removing light was conducted at 80oC. I suggest the authors add a control experiment by removing light only, and conduct the reaction at 0oC. Meanwhile, when air is used, the conversion reaches 100%, with 13% yield of the products. I suggest the authors add comments on this point.

2. In addition to the yields, the conversion of each reaction system should be added. According to the results in Figure 3, modest yields (50-70%) were observed in certain reaction systems. What about the selectivity of these reactions? Only the results on stereoselectivity are given in the article, but the results on regioselectivity should also be taken into account. Meanwhile, when discussing on the diastereoselectivity, syn-, anti- are more reasonable than the cis-, trans- descriptions.

Reviewer #2:

Remarks to the Author:

The present manuscript describes Ag₃PO₄ NPs-promoted visible-light pericyclic reactions. Surface on the Ag₃PO₄ NPs can adsorb electron-rich aromatic alkenes. Photoirradiation of the Ag₃PO₄ NPs can generate long-lived cationic radical species of aromatic alkenes. These are key species for successful pericyclic reactions. In this reviewer's opinion, the work seems too specialized. Model reactions do not contain elements of novelty very much. DFT simulations might give important information, but they look preliminary. This reviewer does not recommend publication in high-impact Nature Communications.

There are a few points to be improved.

- "Styrene" indicates the chemical compound. Thus, the authors should not use "styrene" instead of aromatic alkene.
- This reviewer thought the nanoparticle structure is more important than bulk and solid Ag₃PO₄. Structural character might be the center of the present work. But the nanostructure is not simulated by DFT calculations. DFT calculation should be used for nanostructure simulations.
- In Figure 4a, how were transient absorption spectra obtained? The authors should indicate solution or solid clearly. If the spectra were obtained in the liquid phase, it suggests that the radical cationic species exists in the liquid phase.
- In Figure 7, steps II and III suggest that Ag₃PO₄ is excited, then electron transfer occurs. But the adsorbed aromatic alkene on the surface might undergo the excitation directly. The authors should discuss which species can accept light energy.
- In Figure 7, the final step should be the electron transfer from the reduced Ag₃PO₄ to the cationic radical product. The step should be discussed more.

Reviewer #3:

Remarks to the Author:

Yifeng Wang and co-workers developed the efficient semiconductor photocatalysis using Ag₃PO₄ nanoparticles for intramolecular and intermolecular radical cation-mediated cycloadditions. Wang group have developed various heterogeneous photocatalysis with TiO₂ and Ag/Ag₃PO₄ systems, to generate styrene radical cation species, and successfully described the oxidative dimerization for

aryltetralone synthesis. In two previous reports, they reported that TiO₂ and Ag/Ag₃PO₄ selectively oxidize styrenes with high efficiency under visible light irradiation, and the photocatalyst could be facilely recycled five times with a slight decrease in activity.

This manuscript discloses the single electron oxidation strategy of anethole derivatives, beta-substituted styrenes, to generate the anethole radical cation species and afford the [2+2] dimerization reaction and [4+2] Diels-Alder reaction. They emphasized that anethole radical cation on the NP's surfaces exhibited a 10⁸ times longer lifetime than one in the homogeneous systems. However, the synthetic approaches for anethole radical cation-mediated cycloaddition have been widely investigated employing single electron oxidants, several well-known Ir/Ru photocatalysts and organophotocatalysts, therefore it would be much more magnificent if the new reactions were found which are difficult to afford by traditional photocatalysis. And in terms of synthetic view, the logic of enabling a more efficient reaction by extending the lifetime of the radical cation intermediate seems rather weak, because the substrate anethole is more easily oxidized than terminal styrene. It is also difficult to say that the reaction efficiency could be dramatically increased, when 8~20 mol% of Ag₃PO₄ is used and recycled 5 times, compared to the use of 0.5 ~ 1 mol% of photocatalyst in the previous reports.

Although the mechanistic examination and analysis are well conducted and the practical examples are impressive, this reviewer is not convinced that this work is of sufficient novelty and importance to warrant publication in Nat. Commun, considering the recent high scientific level of Nat. Commun journals. I would like to recommend a specific journal that deals with catalyst, inorganic, organic, or green chemistry.

1. In line 83, Ag/Ag₃PO₄ system afforded a substantially lower yield of ~ 60%, and it was described that Ag NPs are not the photocatalysts, and Ag₃PO₄ is the true photocatalyst. Can we conclude that no photocatalysis has occurred just with a low reaction yield? As explained in line 69 that different Ag₃PO₄ samples exhibited in distinct reaction rates and hardly affected final product yields, the composition and shape of Ag NPs and reaction condition affect the reaction efficiency?
2. For similar reasons, a description of the preparation method or shape of the other catalysts compared in figure 2d needs to be required. Comparing the general inorganic silver salts (AgCl, AgBr, and AgI) with self-synthesized Ag₃PO₄ NP seems to have a big difference.
3. Some relevant reports supported the radical chain mechanism in the [2+2] and [4+2] cycloaddition reaction. How can we be sure about the Ag₃PO₄ mechanism in Figure 7?
4. Compounds 6d, 6e, 6f, 6n, and 6r appear to contain impurities based on the spectra and this casts some questions over the reported yields. Compounds 2d, 4e, 6m, 6o, 8g and 8h; Fluorine coupling is not accounted for in the ¹³C tabulation, and please report also the ¹⁹F-NMR analysis.

Reviewer #4:

Remarks to the Author:

In this work, Wang and coworkers introduce Ag₃PO₄ semiconductor nanoparticles as a visible light heterogenous photocatalyst for radical cation-mediated cycloaddition reactions. As the CB electrons of Ag₃PO₄ are weakly reducing, these nanoparticles have reduced issues with back-electron transfer to the substrate radical cation, a common issue that inhibits the photocatalytic efficiency of other semiconductors. A nice substrate scope, and detailed mechanistic studies are presented. In particular, the transient absorption data is really compelling, demonstrating an increase in the lifetime of the anethole radical cation to 1900 us.

I believe that once the authors address my comments and concerns listed below, this manuscript would be well suited for publication in Nature Communications.

Comment 1: Please add the wattage of the light sources used, and the distance from the light source to the sample to the supporting information to facilitate reproducibility.

Comment 2: On lines 69 & 70, the authors indicate that different Ag₃PO₄ samples exhibited different reaction rates but did not affect overall product yields. I could not find these data in the manuscript or SI. Perhaps they could be added to Table S1? It would be useful to demonstrate that commercial

Ag₃PO₄ works comparably, for practitioners who do not wish to synthesize the catalyst.

Comment 3: I do not agree with using "mol %" for the catalyst when the reactions are heterogeneous. It is difficult to interpret mol% when it is not clear with regards to how many active sites your material possesses. I would recommend either simply stating how many mg of catalyst was used or using "mg/mL of solvent". I find "mg/mL" to be the most useful, as in my experience, this descriptor is the most applicable when trying to scale heterogeneous photochemical reactions.

Comment 4: The authors should consider adding the control reactions to Scheme 1, as I believe that would be beneficial to the readers.

Comment 5: In lines 99 and 100, the authors indicate that electron accumulation on TiO₂ rendered the catalyst inert once a conversion of 18% was reached. Typically, TiO₂ photocatalyzed reactions that utilize photogenerated holes on the TiO₂ surface are not performed in a degassed environment, as the oxygen dissolved in solution quenches the CB electrons, slowing down electron-hole recombination. Perhaps a fairer comparison would be to also check the reactivity of TiO₂ under an air atmosphere?

Comment 6: On line 135, the authors indicate the gram scale reaction was performed at ambient temperatures (1-10 °C). Surely this is a typo? Ambient temperatures are generally 20-25 °C, therefore a room this cold would be very atypical.

Comment 7: Regarding the gram scale reaction, the procedure seems to be missing from the SI. Please add the procedure to the SI.

Comment 8: Just a general comment: while it is nice to say the reaction was mediated by sunlight, I don't find this to be especially meaningful. Sunlight is a difficult thing to reproduce day to day, location to location. I do appreciate the authors have included the sunlight intensity in their manuscript, however. The implementation of sunlight in organic synthesis is not something that is being considered industrially, especially with the advances in LED technology. In my opinion, it would be more meaningful if the authors performed the gram scale reaction with a LED of a specified wattage, as this simplifies reproducibility in other labs.

Comment 9: While the transient absorption data indicates a lifetime for the anethole radical cation of 1900 us, the authors indicate that it has a lifetime of "several hours" based on the results from Figure 4d. An alternative explanation to the continued conversion of the stirred sample is that there are still photogenerated holes on the Ag₃PO₄ that can react with anethole, not that the anethole radical cation continues to persist for several hours. An increase in the lifetime from 25 us to 1900 us is still of note, but I would caution the authors for claiming that the radical cation itself lives for hours, as the transient absorption data does not correlate with this statement.

Point-to-point response

Referees' comments are listed below in black, followed by authors' point-to-point responses in blue text. The changes are summarized following the response and highlighted in the revised manuscript for easy identification.

Reviewer #1

In this paper, Wang and co-workers reported a visible light driven [2+2] and [4+2] reactions mediated by Ag_3PO_4 nanoparticles. The reactions show advantages in the mild and additive free reaction conditions, scalable synthesis, and the ease in catalyst regeneration. The paper is well written, and the discussions are reasonable. I have only a few comments:

Response: We thank the anonymous reviewers for careful reading and constructive comments, which have strengthened this work.

Q1. The effect of temperature and air should be examined more carefully. 0 °C was used in the standard condition, but the control experiments by removing light was conducted at 80 °C. I suggest the authors add a control experiment by removing light only, and conduct the reaction at 0 °C. Meanwhile, when air is used, the conversion reaches 100%, with 13% yield of the products. I suggest the authors add comments on this point.

Response:

1. We report the results of the dark reaction at 80 °C because the reaction rate typically increases with temperature. This is typical for the majority of thermal reaction systems. If Ag_3PO_4 cannot initiate direct oxidation of **1a** at high temperatures, it cannot trigger the same reaction at 0 °C. Therefore, we did not include the results of the dark reaction at 0 °C previously.
2. In air atmosphere, Ag_3PO_4 can lead to over-oxidation of **1a** to 4-methoxybenzaldehyde, with almost no cycloaddition products generated.

Changes made:

1. We have changed the following description on line 80: "The control experiments demonstrated that in the absence of Ag_3PO_4 or light, no conversion of **1a** occurred (Table S1)."
2. We added the result of the dark reaction at 0 °C to Table S1.
3. We have added the following description to line 82: "Under air atmosphere, the conversion

of **1a** could reach 100%, but 4-methoxybenzaldehyde was the main product (82% yield). The yield of the cycloaddition product **2a** was only 13%.”

Q2. In addition to the yields, the conversion of each reaction system should be added. According to the results in Figure 3, modest yields (50-70%) were observed in certain reaction systems. What about the selectivity of these reactions? Only the results on stereoselectivity are given in the article, but the results on regioselectivity should also be taken into account. Meanwhile, when discussing on the diastereoselectivity, syn-, anti- are more reasonable than the cis-, trans- descriptions.

Response:

We appreciate the reviewer's comments.

1. In numerous papers on organic synthesis, the conversion values of the substrates are not reported because the products are separated by column chromatography, and the yields of the products are calculated by weighing on a balance. However, few people recover the raw materials through column chromatography, making it impossible to know how much substrates are left. In order to answer the yield question raised by the reviewer, we re-measured the conversion and yield of some representative examples by using ^1H NMR. We included the results and discussions in the SI.

2. For regioselectivity, there are two products of intermolecular [2+2] cycloaddition: head-to-head and head-to-tail. They can be distinguished by ^1H NMR analysis. No head-to-tail products were generated in our study. There is only one product for intramolecular [2+2] cycloadditions, and there is no regioselectivity issue.

Theoretically, there are two products for the [4+2] cycloaddition between **2a** and asymmetric butylenes. Namely, products **8b**, **8c**, **8d**, **8f**, and **8h** should have their isomers. However, in ^1H NMR analysis, we found that these products were very pure, and their spectra were consistent with literature reports without any isomers, indicating that the regioselectivity should be 100%. This is because the groups possessing large steric hindrance on the butadiene side chain tend to position themselves distantly from the p-methoxyphenyl group in the resultant [4+2] product to reduce steric hindrance. On the other hand, the other products (**8a**, **8e**, and **8g**) do not have regioselectivity issues.

3. We agree with the reviewer's opinion about the use of “syn-” and “anti-”.

Changes made:

1. We have added the following description to line 124: “The corresponding intermolecular [2+2] products (**2a–2h** and **4a–4i**) were obtained in high yields and high regioselectivity (head-to-head; see Table S4 for ¹H NMR analysis).”

2. We have added the following description to line 136: “The products **8b, 8c, 8d, 8f, and 8h** were isolated in high regioselectivity without isomers. This can be attributed to the large steric hindrance of the butadiene side chain (see Table S4 for ¹H NMR analysis).” .

4. We added the ¹H NMR comparison of the products (**2a, 4a** and **8b**) as Table S4, followed by the discussion on the isomers of [2+2] and [4+2] products.

Reviewer #2

The present manuscript describes Ag₃PO₄ NPs-promoted visible-light pericyclic reactions. Surface on the Ag₃PO₄ NPs can adsorb electron-rich aromatic alkenes. Photoirradiation of the Ag₃PO₄ NPs can generate long-lived cationic radical species of aromatic alkenes. These are key species for successful pericyclic reactions. In this reviewer’s opinion, the work seems too specialized. Model reactions do not contain elements of novelty very much. DFT simulations might give important information, but they look preliminary. This reviewer does not recommend publication in high-impact Nature Communications.

Response:

Thank the reviewer for his comments and suggestions.

As can be seen, review #2 has comprehended the following key points of this article: (1) Using Ag₃PO₄ semiconductor NPs to achieve visible-light catalyzed cycloaddition, and (2) Photoirradiation of the Ag₃PO₄ NPs can generate long-lived cationic radical species of aromatic alkenes.

Herein, we want to persuade review #2 further to trust the innovation and importance of this article based on the following points:

1. This article marks the inaugural application of Ag₃PO₄, an inorganic semiconductor and visible-light photocatalyst, for pericyclic reactions.

2. The pericyclic reactions have broad interests. The radical pericyclic reaction of olefins holds

importance due to its ability to produce a wide range of crucial fine chemicals. Furthermore, it continues to be a subject of active research and remains an area of significant interest. Our study covers common pericyclic reaction types, including intramolecular [2+2] cycloaddition, intermolecular homo [2+2] cycloaddition, intermolecular cross [2+2] cycloaddition, and [4+2] cycloaddition. It is expected that further advancements in radical cation pericyclic reactions, as well as the exploration of reactions beyond pericyclic reactions, will be achieved in accordance with the current system's principles.

3. This article proposes a new mechanism for radical cation reactions occurring at interfaces of semiconductors. Extending the lifetime of radical cation intermediates of olefins and regulating their reaction activity is crucial for achieving pericyclic reactions. In recent years, a few innovative studies have attempted to use various strategies to stabilize radical cations, such as refs. 17, 18, and 21. Here, we used nanoparticles to stabilize radical cations and regulate their reactivity. This is the first time this phenomenon has been observed in a heterogeneous system, and we have successfully extended the lifespan of $1a^{*+}$ to >2 ms, which is striking. This mechanism will be inspiring for similar systems and may also be extended to radical anion reaction systems.

4. Compared to recent state-of-the-art studies, the advantage of using Ag_3PO_4 is also significant, in terms of mild and additive-free reaction conditions, scalable synthesis, and ease in catalyst regeneration.

5. Photocatalytic organic synthesis has been a rapidly developing research direction in the past 15 years. We believe that this article will attract the attention of researchers from multiple fields. Therefore, we disagree with the viewpoint of reviewer 2 that the work is “too specialized”.

There are a few points to be improved.

Q1. “Styrene” indicates the chemical compound. Thus, the authors should not use “styrene” instead of aromatic alkene.

Response and changes made:

We appreciate the reviewer’s advice.

During revision, we replaced “styrene” with “aromatic alkene”.

Q2. This reviewer thought the nanoparticle structure is more important than bulk and solid Ag_3PO_4 . Structural character might be the center of the present work. But the nanostructure is not simulated by DFT calculations. DFT calculation should be used for nanostructure simulations.

Response:

1. The above questions may be caused by the misunderstanding of reviewer #2 due to our frequently use of Ag_3PO_4 nanoparticles in the manuscript, especially in the title, abstract, and conclusion. Here, we need to clarify that our catalyst sample itself is NPs, so using nanoparticles is only to describe objective facts. We did not intend to emphasize that Ag_3PO_4 nanoparticles have better performance than bulk and solid Ag_3PO_4 .

2. This article did not focus on the structure and properties of Ag_3PO_4 , nor did it study the structure-activity relationship. We used several types of Ag_3PO_4 nanoparticles, such as cubic, rhombic, dodecahedral, and spherical, to demonstrate that the structure of Ag_3PO_4 changes the initial rate by affecting adsorption.

3. It is challenging to simulate NPs using DFT while simulating NPs with both anions and cations is even more difficult. This is because we cannot know the exact structure of NPs (such as defect distribution). If the model has too few atoms, it cannot simulate the actual structure of an NP, while if the number of atoms exceeds 100 (such as a $(\text{Ag}_3\text{PO}_4)_{13}$ NP), the computational resources required are too large to bear. We did not find a reference on the DFT simulation of Ag_3PO_4 NPs, and establishing a model of Ag_3PO_4 NPs from scratch and conducting related research should be a full study. Such theoretical simulation is clearly beyond the scope of this study, which focuses on experimental results.

4. The current DFT calculation only calculates the adsorption energies of **1a** substrate and **2a** product, as well as the Bader charge of **1a** upon adsorption. Using a two-dimensional plane model is reasonable because (1) the diameter of NPs is about 230 nm, the surface curvature of each NP should be very small, and (2) the size of each **1a** or **2a** molecule is equivalent to the length of two phosphate anions. Therefore, the Ag_3PO_4 NP surface can be considered an infinite plane for **1a** and **2a**.

Changes made:

1. We have removed the term “nanoparticles” from many places in the revised manuscript.

2. We added the following description of DFT simulation to line 238 in the main text: “Given the substantial dimensions of the Ag_3PO_4 particles and the comparable sizes of **1a** or **2a** molecules to only a few phosphate anions, it is reasonable to regard the surface of Ag_3PO_4 particles as an infinitely expansive plane for adsorbing **1a** and **2a**.”

Q3. In Figure 4a, how were transient absorption spectra obtained? The authors should indicate a solution or solid clearly. If the spectra were obtained in the liquid phase, it suggests that the radical cationic species exists in the liquid phase.

Response:

We appreciate the reviewer’s detailed advice.

1. The first question might be a result of misunderstanding. The following sentence, “The transmittance LFP spectrum of a **1a**/ Ag_3PO_4 /HFIP suspension is depicted in Figure 4a” indicates the transient spectra were measured in transmittance mode. The experimental section provides a detailed description of the procedure of transient absorption spectroscopy. We have rewritten the relevant content in the revised manuscript to clarify the experimental procedure.

2. Reviewer # 2 may have doubts about the interpretation of our LFP result. The signals of **1a**^{•+} were still detected at 1900 μs in the LFP spectra. This can be explained in two ways: (a) the measured signal is attributed to **1a**^{•+} located on the surface of Ag_3PO_4 NPs, or (b) the measured signal is **1a**^{•+} in the solution bulk which is slowly and continuously released from the surface of Ag_3PO_4 . Explanation (b) may be the scenario envisioned by Reviewer #2. However, explanation (b) can be dismissed for the following reasons. According to this assumption, the signal intensity should be proportional to the desorption rate of **1a**^{•+}. In contrast, the signal of **1a**^{•+} shows minimal decay after 500 μs , suggesting that the desorption rate remains constant. This contradicts the widely accepted Langmuir desorption kinetics, which states that the desorption rate is proportional to the coverage. In addition, ultrasonic treatment can make Ag_3PO_4 highly dispersed in HFIP solvent, making the suspension appear homogeneous and transparent. This may be the main reason why the **1a**^{•+} adsorbed on Ag_3PO_4 can be measured in transmission mode.

Unambiguously, neither explanation will affect one of the main conclusions of this article, namely, that **1a**^{•+} have an ultra-long lifetime.

Changes made:

1. To clarify that the LFP spectra were obtained in transmission mode, a comprehensive description was added to the experimental section of the SI. Meanwhile, line 166 of the main text is rewritten as follows: “The LFP transient absorption spectra of the **1a**/Ag₃PO₄/HFIP suspension were obtained in transmission mode. The representative spectra are depicted in Figure 4a.”

2. To further confirm that the signal detected by LFP is **1a**⁺⁺ from the surface of Ag₃PO₄ NPs, the line 197 of the main text is rewritten as follows: “This verifies that the signals originate from **1a**⁺⁺ adsorbed on the Ag₃PO₄ surfaces rather than in the solution bulk.”

3. We added the following discussion to SI:

The signals of **1a**⁺⁺ can be detected after 1900 μs by LFP spectra. This can be explained in two ways: (a) the long-lived signal is attributed to **1a**⁺⁺ located on the surface of Ag₃PO₄ NPs, or (b) the persistent signal of **1a**⁺⁺ is attributed to the **1a**⁺⁺ in the solution bulk, which is slowly and continuously released from the surface of Ag₃PO₄.

However, explanation (b) can be ruled out for the following reasons. According to this assumption, the signal intensity should be equivalent to the desorption rate of **1a**⁺⁺. Because the signal of **1a**⁺⁺ almost no longer decays after 500 μs, the unchanged signal intensity indicates that the desorption rate is constant. This contradicts the widely accepted Langmuir desorption kinetics, which states that the desorption rate is proportional to the coverage. In addition, ultrasonic treatment can make Ag₃PO₄ highly dispersed in HFIP solvent, making the suspension appear quite homogeneous and transparent. This may be the main reason that the signal of **1a**⁺⁺ adsorbed on the surface of Ag₃PO₄ can be measured in transmission mode.

Unambiguously, neither explanation will affect one of the main conclusions of this article, namely, that **1a**⁺⁺ has an ultra-long lifetime on the Ag₃PO₄ surfaces.

Q4. In Figure 7, steps II and III suggest that Ag₃PO₄ is excited, then electron transfer occurs. But the adsorbed aromatic alkene on the surface might undergo the excitation directly. The authors should discuss which species can accept light energy.

Response:

We appreciate the reviewer’s question. This is a very important question that we have already

addressed.

1. In line 80 of the main text, we pointed out that the conversion of **1a** does not occur without Ag_3PO_4 . This evidence substantiates the involvement of Ag_3PO_4 as the photocatalyst.

2. During revision, we measured the UV-vis spectrum of **1a**. **1a** can only absorb light below 320 nm. In contrast, Figure 2b shows that the absorption edge of Ag_3PO_4 is approximately 510 nm. Figure 2b also shows that Ag_3PO_4 can effectively harvest visible light up to 500 nm to initiate the **1a** \rightarrow **2a** cycloaddition, which is close to its absorption edge. From the above three points, we can firmly draw the conclusion that the [2+2] cycloaddition reaction is initiated by photoexcitation of Ag_3PO_4 .

Changes made:

1. We added a UV-vis spectrum of **1a** to SI.

Figure S3. The UV-Vis spectra of **1a** in MeCN.

2. We added the following description to line 88, in the main text: “The UV-vis absorption spectrum of **1a** shows it can only absorb light below 320 nm, and only Ag_3PO_4 can be excited by the light source (Figure S3).”

Q5. In Figure 7, the final step should be the electron transfer from the reduced Ag_3PO_4 to the cationic radical product. The step should be discussed more.

Response:

We appreciate the insightful comment.

Firstly, we agree with reviewer #2's viewpoint that the final step is the electron transfer from the reduced Ag_3PO_4 to the cationic radical product.

Secondly, the conduction band potential of Ag_3PO_4 is +0.45 V vs. NHE. The redox potential of $2\mathbf{a}^+/2\mathbf{a}$ is approximately +1.5 V vs. NHE. From a thermodynamic perspective, the transfer of an electron from the reduced Ag_3PO_4 to $2\mathbf{a}^+$ is an exothermic reaction, with a high driving force. Moreover, step V is not rate-limiting, as has been discussed in the main text and SI. We did not delve into this step in our previous manuscript because we wanted to focus more on the rate-limiting step.

Changes made:

In the following paragraph of Figure 7, We added the following description: “The final step (step V) is the ring closure reaction of $2\mathbf{a}^+$. The potential of $2\mathbf{a}^+/2\mathbf{a}$ is approximately +1.5 V vs. NHE, much higher than the E_{CB} of Ag_3PO_4 . Thermodynamically, the transfer of an electron from the reduced Ag_3PO_4 to $2\mathbf{a}^+$ is an exothermic reaction.”

Reviewer #3

Yifeng Wang and co-workers developed the efficient semiconductor photocatalysis using Ag_3PO_4 nanoparticles for intramolecular and intermolecular radical cation-mediated cycloadditions. Wang group have developed various heterogeneous photocatalysis with TiO_2 and $\text{Ag}/\text{Ag}_3\text{PO}_4$ systems, to generate styrene radical cation species, and successfully described the oxidative dimerization for aryltetralone synthesis. In two previous reports, they reported that TiO_2 and $\text{Ag}/\text{Ag}_3\text{PO}_4$ selectively oxidize styrenes with high efficiency under visible light irradiation, and the photocatalyst could be facilely recycled five times with a slight decrease in activity.

This manuscript discloses the single electron oxidation strategy of anethole derivatives, beta-substituted styrenes, to generate the anethole radical cation species and afford the [2+2] dimerization reaction and [4+2] Diels-Alder reaction. They emphasized that anethole radical cation on the NP's surfaces exhibited 10^8 times longer lifetime than one in the homogeneous systems. However, the synthetic approaches for anethole radical cation-mediated cycloaddition have been widely investigated employing single electron oxidants, several well-known Ir/Ru photocatalysts and organophotocatalysts, therefore it would be much more magnificent if the new reactions were found which are difficult to afford by traditional photocatalysis. And in terms of synthetic view, the logic of enabling a more efficient reaction by extending the lifetime

of the radical cation intermediate seems rather weak, because the substrate anethole is more easily oxidized than terminal styrene. It is also difficult to say that the reaction efficiency could be dramatically increased, when 8~20 mol% of Ag_3PO_4 is used and recycled 5 times, compared to the use of 0.5 ~ 1 mol% of photocatalyst in the previous reports.

Although the mechanistic examination and analysis are well conducted and the practical examples are impressive, this reviewer is not convinced that this work is of sufficient novelty and importance to warrant publication in Nat. Commun, considering the recent high scientific level of Nat. Commun journals. I would like to recommend a specific journal that deals with catalyst, inorganic, organic, or green chemistry.

Response:

We express our gratitude to the reviewer 3 for his careful examination and valuable feedback.

We would also like to emphasize the following points:

1. Reviewer #3 may overlook the significance and difficulties associated with employing semiconducting, heterogeneous, and recyclable visible-light photocatalysts to accomplish pericyclic reactions.
2. We respectfully hold a differing viewpoint from reviewer #3 regarding his assessment of the "relatively weak extension of the lifetime of the radical cations intermediate." The observation of this phenomenon is indeed remarkable, as highlighted by Reviewer #4. Furthermore, in recent years, several researchers have utilized diverse sophisticated methods to control the lifespan and reactivity of $\mathbf{1a}^{+\bullet}$, as indicated by refs. 17, 18, and 21. Specifically, they utilized large organic and inorganic ions in their experiments; however, these species lack recyclability. Our work exhibits a notable degree of innovation in comparison to theirs.
3. When considering the outcome, Table S5-S7 comprehensively compares our system with the reported ones. Besides, using Ag_3PO_4 offers several advantages in terms of product separation, scale-up synthesis, and catalyst reuse, which were challenging to achieve in previous homogeneous systems. Koenig et al. noted that "the use of Ir/Ru photocatalyst is restricted on account of incompatibility with strong nucleophiles, electrophiles, or reactive radical intermediates, leading eventually to catalyst deactivation"(ref 24). Thus, it has been generally known that Ir/Ru photocatalysts exhibit limited stability when subjected to turnover conditions.
4. Our findings highlight the potential of inorganic semiconductors in the challenging radical

cation/anion-mediated synthesis driven by sunlight. The new mechanism may inspire new ideas for more challenging radical cation/anion-mediated solar synthesis using inorganic semiconductor photocatalysts.

Q1. In line 83, Ag/Ag₃PO₄ system afforded a substantially lower yield of ~ 60%, and it was described that Ag NPs are not the photocatalysts, and Ag₃PO₄ is the true photocatalyst. Can we conclude that no photocatalysis has occurred just with a low reaction yield? As explained in line 69 that different Ag₃PO₄ samples exhibited in distinct reaction rates and hardly affected final product yields, the composition and shape of Ag NPs and reaction condition affect the reaction efficiency?

Response:

The comments are valuable and thoughtful.

1. Ag₃PO₄ inevitably generates AgNPs under irradiation. AgNPs are a common surface plasmon resonance (SPR) material that strongly absorbs visible light. The reason we examined the catalytic activity of Ag/Ag₃PO₄ is to determine what is the real light absorber for the reaction. Indeed, we found that AgNPs inhibited the reaction, indicating that AgNPs do not contribute to the [2+2] cycloaddition.

We agree with Reviewer #3's viewpoint that “no photocatalysis has occurred just with a low reaction yield”, that is, all **2a** is the product of photocatalysis.

2. We feel that reviewer #3 demonstrates a strong inclination towards the field of material sciences related to AgNPs/Ag₃PO₄ material, and a particular interest in the role of AgNPs. However, we believe these are not our study's primary focus. Our so-called “different Ag₃PO₄ samples” refers to Ag₃PO₄ samples that initially did not contain AgNPs at all. We know that AgNPs/Ag₃PO₄ composite materials are commonly used photocatalysts, with many reports, such as reference 40. However, based on the fact that Ag₃PO₄ is a photocatalyst in this article and AgNPs are not, the presence of AgNPs will have at least two effects: (1) covering the surface sites of Ag₃PO₄, inhibiting the interaction between **1a** and the surface of Ag₃PO₄; (2) photoexcited hot electrons on AgNPs may reduce **1a**⁺, thereby inhibiting the formation of **2a**. To answer reviewer #3's question: “the composition and shape of AgNPs and reaction condition affect the reaction efficiency”, we carried out supplementary experiments and added the results

and discussion to the revised manuscript.

Changes made:

1. In the main text line 92, we added the following description: “..., all the **2a** are products of Ag_3PO_4 photocatalysis”.
2. In the main text line 90, the following results and discussion are added: “Furthermore, as the loading of AgNPs increased, the final yield of **2a** decreased (Figures S4-S6).”
3. The preparation method, reaction kinetics, basic sample characterization data (UV-vis spectra, SEM, elemental analysis) are added to SI as Figure S4-S6.

Q2. For similar reasons, a description of the preparation method or shape of the other catalysts compared in figure 2d needs to be required. Comparing the general inorganic silver salts (AgCl, AgBr, and AgI) with self-synthesized Ag_3PO_4 NP seems to have a big difference.

Response:

We appreciate the reviewer' detailed suggestion.

The preparation method and basic characterization data of the samples in Figure 2d have been added to the revised SI. Indeed, compared to the self-synthesized Ag_3PO_4 , general inorganic silver salts (AgCl, AgBr, and AgI) not only lead to excessive oxidation of **1a** and low yield of **2a**, but they also quickly became inactive. We speculate that Reviewer #3 is concerned about the activities of the self-synthesized AgCl, AgBr, and AgI. In fact, they are very similar to commercial AgCl, AgBr, and AgI.

Changes made:

1. We have added the following description to line 104:
“As shown in Figure 2d, when the other silver salts, such as AgCl, AgBr, and AgI, were used, the conversion of **1a** and the selectivity of **2a** were significantly lower than that of Ag_3PO_4 . Both commercially available and self-synthesized AgCl, AgBr, and AgI decomposed rapidly upon irradiation (see Figure S8 for characterization).”
2. The procedures for synthesizing Ag_3PO_4 with different morphologies and the methods for synthesizing AgCl, AgBr, AgI, and other catalysts were added to SI.

Q3. Some relevant reports supported the radical chain mechanism in the [2+2] and [4+2]

cycloaddition reaction. How can we be sure about the Ag_3PO_4 mechanism in Figure 7?

Response: This is a very good question.

We found it difficult to rule out the contribution of the radical chain mechanism to the reactions. However, the conduction band potential of Ag_3PO_4 is +0.45 V vs. NHE. The potential of the $2\mathbf{a}^+/2\mathbf{a}$ pair is +1.5 V vs. NHE. The potential of the $1\mathbf{a}^+/1\mathbf{a}$ pair is +1.3 V vs. NHE. From a thermodynamic perspective, the electron transfer from the reduced Ag_3PO_4 to $2\mathbf{a}^+$ has a greater thermodynamic driving force than that from $1\mathbf{a}$ to $2\mathbf{a}^+$. Adsorption of $2\mathbf{a}^+$ on the photo-excited Ag_3PO_4 surfaces should also facilitate the reduction of $2\mathbf{a}^+$. Therefore, achieving step V through conduction band electrons should be easier than through radical chain reactions.

Changes made:

In line 292, we have added the following description on step V:

“Step V may also occur through a radical chain mechanism.²¹ Considering the potential of $1\mathbf{a}^+/1\mathbf{a}$ which is approximately +1.3 V vs. NHE, the ring closure of $2\mathbf{a}^+$ by e_{CB}^- exhibits a much larger driving force than that by the radical chain process.”

Q4. Compounds **6d**, **6e**, **6f**, **6n**, and **6r** appear to contain impurities based on the spectra and this casts some questions over the reported yields. Compounds **2d**, **4e**, **6j**, **6m**, **6o**, **8g** and **8h**; Fluorine coupling is not accounted for in the ^{13}C tabulation, and please report also the ^{19}F -NMR analysis.

Response:

We appreciate the reviewer's advice.

During the revision, we re-performed the catalytic reactions and obtained neat spectra of the compounds mentioned above.

Changes made:

1. We have carefully recalculated the isolated yields.
2. We updated the corresponding NMR spectra in the SI. Fluorine coupling and ^{19}F -NMR analyses were reported in the SI.

Reviewer #4

In this work, Wang and coworkers introduce Ag_3PO_4 semiconductor nanoparticles as a visible light heterogenous photocatalyst for radical cation-mediated cycloaddition reactions. As the CB

electrons of Ag_3PO_4 are weakly reducing, these nanoparticles have reduced issues with back-electron transfer to the substrate radical cation, a common issue that inhibits the photocatalytic efficiency of other semiconductors. A nice substrate scope, and detailed mechanistic studies are presented. In particular, the transient absorption data is really compelling, demonstrating an increase in the lifetime of the anethole radical cation to 1900 μs .

I believe that once the authors address my comments and concerns listed below, this manuscript would be well suited for publication in Nature Communications.

Q1. Please add the wattage of the light sources used, and the distance from the light source to the sample to the supporting information to facilitate reproducibility.

Response:

Thanks for the reviewer's valuable and detailed suggestions.

Changes made:

The wattage of the light sources and the following information have been added to the revised SI.

Figure S19. The picture of light sources.

Q2. On lines 69 & 70, the authors indicate that different Ag_3PO_4 samples exhibited different reaction rates but did not affect overall product yields. I could not find these data in the manuscript or SI. Perhaps they could be added to Table S1? It would be useful to demonstrate that commercial Ag_3PO_4 works comparably, for practitioners who do not wish to synthesize the catalyst.

Response:

Thanks for bringing up the suggestion.

Changes made:

1. The outcome of commercial Ag_3PO_4 and different Ag_3PO_4 samples was added to Table S1.
2. We added corresponding description in line 75: “Due to this equilibrium, the 82% yield was also achieved using various Ag_3PO_4 samples, including a commercially available Ag_3PO_4 and several self-synthesized faceted Ag_3PO_4 nanocrystal samples (see Table S1 for screening of the experimental conditions). This implies that the catalyst is easily accessible.”

Q3. I do not agree with using “mol %” for the catalyst when the reactions are heterogeneous. It is difficult to interpret mol% when it is not clear with regards to how many active sites your material possesses. I would recommend either simply stating how many mg of catalyst was used or using “mg/mL of solvent”. I find “mg/mL” to be the most useful, as in my experience, this descriptor is the most applicable when trying to scale heterogeneous photochemical reactions.

Response:

We appreciate the advice. “mg/mL” is more suitable than “mol %”.

Changes made:

We have used mg mL^{-1} in main text and SI.

Q4. The authors should consider adding the control reactions to Scheme 1, as I believe that would be beneficial to the readers.

Response:

We agree with you.

Changes made:

In line 84, we added the corresponding description: “We also performed the reaction in other solvents like tetrahydrofuran (THF), MeOH, EtOAc, and $\text{CF}_3\text{CH}_2\text{OH}$, but they were not as effective as HFIP.”

Q5. In lines 99 and 100, the authors indicate that electron accumulation on TiO_2 rendered the catalyst inert once a conversion of 18% was reached. Typically, TiO_2 photocatalyzed reactions that utilize photogenerated holes on the TiO_2 surface are not performed in a degassed environment, as the oxygen dissolved in the solution quenches the CB electrons, slowing down

electron-hole recombination. Perhaps a fairer comparison would be to also check the reactivity of TiO₂ under an air atmosphere?

Response:

We appreciate the suggestion. We studied the activity of TiO₂ under air atmosphere.

Changes made:

In line 109, we added the following description: “When TiO₂ was used under air atmosphere, the accumulated electrons can be efficiently quenched by O₂. However, in this case, the yield of **2a** was less than 10% due to the over-oxidization of **1a** to 4-methoxybenzaldehyde (67%).”

Q6. On line 135, the authors indicate the gram scale reaction was performed at ambient temperatures (1-10 °C). Surely this is a typo? Ambient temperatures are generally 20-25 °C, therefore a room this cold would be very atypical.

Response:

Thanks for this comment. This is because our experiment was conducted outside of the building in the winter when the temperature ranged from 1 to 10 °C. Summer temperatures can reach 38 °C, but we didn't perform experiment in the summer.

Changes made:

We changed “ambient temperature” to “outdoor temperature”.

Q7. Regarding the gram scale reaction, the procedure seems to be missing from the SI. Please add the procedure to the SI.

Changes made:

General procedure of gram scale reaction has been added to the SI.

Q8. Just a general comment: while it is nice to say the reaction was mediated by sunlight, I don't find this to be especially meaningful. Sunlight is a difficult thing to reproduce day to day, location to location. I do appreciate the authors have included the sunlight intensity in their manuscript, however. The implementation of sunlight in organic synthesis is not something that is being considered industrially, especially with the advances in LED technology. In my opinion, it would be more meaningful if the authors performed the gram scale reaction with a LED of a

specified wattage, as this simplifies reproducibility in other labs.

Response:

We agree with the description. We also carried out the gram scale reaction with an LED, and a similar yield was obtained.

Changes made:

We added corresponding description in line 155: “The use of a 425 nm LED lamp achieved yields of **8a** that are comparable to that achieved by sunlight (Table S1). Hence, the Ag₃PO₄/visible light system could facilitate actual industrial production considering the advances in LED technology.”

Q9. While the transient absorption data indicates a lifetime for the anethole radical cation of 1900 μs, the authors indicate that it has a lifetime of “several hours” based on the results from Figure 4d. An alternative explanation to the continued conversion of the stirred sample is that there are still photogenerated holes on the Ag₃PO₄ that can react with anethole, not that the anethole radical cation continues to persist for several hours. An increase in the lifetime from 25 μs to 1900 μs is still of note, but I would caution the authors for claiming that the radical cation itself lives for hours, as the transient absorption data does not correlate with this statement.

Response:

We appreciate the reviewer’s advice.

1. We cannot determine whether the lifetime of photo-generated holes in Ag₃PO₄ can last for several hours. According to the literature reports, the photo-generated holes in semiconductors are transient, while only photo-generated electrons can be long-lived.
2. We accept the reviewer's suggestion to revise the description on the lifetime of **1a^{•+}**.

Changes made:

1. We added the corresponding description in line 187:

“This must not be attributed to the continuous generation of **1a^{•+}** by the photogenerated holes of Ag₃PO₄ after LFP, because it is well-known that the photogenerated holes in semiconductors are transient species. Instead, this indicates that the lifetime of **1a^{•+}** in the **1a**/Ag₃PO₄/HFIP system is more than 1900 μs.”

2. We changed the lifetime of the radical cation from “several hours” to “more than 2 ms” in many places.

Reviewers' Comments:

Reviewer #1:

Remarks to the Author:

The authors have well settled the issues I raised last time, and the paper is acceptable

Reviewer #4:

Remarks to the Author:

[Note from the editor: Reviewer #4 was asked to look over the response given to reviewer #2 and #3 who were unable to assess the revised manuscript.]

I would like to thank the authors for working to address the comments from all the reviewers, and for the detailed response to the reviewers document. I believe the edits have enhanced the manuscript, and I would recommend it for publication in Nature Communications after the following minor comments have been addressed.

Comment 1: In the data tabulations in the SI, there are several compounds that randomly have grey boxes over them. These should be removed.

Comment 2: The authors use the abbreviation "eq" to describe equivalents in the manuscript. Please use the abbreviation "equiv", as "eq" is the abbreviation for equation.

Comment 3: In the introduction, lines 63,64, the authors claim that "this is the first report on employing an isPC for pericyclic reactions under visible-light irradiation. This statement needs to be revised. [2+2] reactions photocatalyzed by CdS under visible light irradiation was first reported by De Mayo in 1986 (Tetrahedron 1986, 42, 6277). More recently, Scaiano and Yoon have reported visible light mediated [4+2] cycloadditions with indoles using TiO₂ (Chem. Commun. 2017, 53, 4335; ACS Catal. 2017, 7, 6440). Weiss and coworkers have also reported [2+2] cycloadditions using quantum dots and visible light irradiation (Nature Chemistry 2019, 11, 1034). I was able to find these after a very short search, and I encourage the authors to more diligently look through the literature when making such statements. These references should be added, and the sentence modified.

Point-to-point response

Referees' comments are listed below in black, followed by authors' point-to-point responses in blue text. The changes are summarized following the response and highlighted in the revised manuscript for easy identification.

Reviewer #1

The authors have well settled the issues I raised last time, and the paper is acceptable.

Response: We thank the anonymous reviewer for their valuable comments and time.

Reviewer #4

I would like to thank the authors for working to address the comments from all the reviewers, and for the detailed response to the reviewers document. I believe the edits have enhanced the manuscript, and I would recommend it for publication in *Nature Communications* after the following minor comments have been addressed.

Response: We thank the anonymous reviewer for careful reading and constructive comments, which have strengthened this work.

Q1. In the data tabulations in the SI, there are several compounds that randomly have grey boxes over them. These should be removed.

Response:

We appreciate the reviewer's advice. We carefully modified the corresponding structural formula during revision.

Q2. The authors use the abbreviation "eq" to describe equivalents in the manuscript. Please use the abbreviation "equiv", as "eq" is the abbreviation for equation.

Response:

We appreciate the reviewer's detailed advice. We have modified the abbreviation in the main text.

Q3. In the introduction, lines 63,64, the others claim that "this is the first report on employing

an isPC for pericyclic reactions under visible-light irradiation. This statement needs to be revised. [2+2] reactions photocatalyzed by CdS under visible light irradiation was first reported by De Mayo in 1986 (Tetrahedron 1986, 42, 6277). More recently, Scaiano and Yoon have reported visible light mediated [4+2] cycloadditions with indoles using TiO₂ (Chem. Commun. 2017, 53, 4335; ACS Catal. 2017, 7, 6440). Weiss and coworkers have also reported [2+2] cycloadditions using quantum dots and visible light irradiation (Nature Chemistry 2019, 11, 1034). I was able to find these after a very short search, and I encourage the authors to more diligently look through the literature when making such statements. These references should be added, and the sentence modified.

Response:

We appreciate the reviewer's suggestion. We have modified the corresponding description and added relevant references.